# Cell Membrane-Cloaked Nanotherapeutics for Targeted Drug Delivery

**DOI:** 10.3390/ijms23042223

**Published:** 2022-02-17

**Authors:** Na-Hyun Lee, Sumin You, Ali Taghizadeh, Mohsen Taghizadeh, Hye Sung Kim

**Affiliations:** 1Institute of Tissue Regeneration Engineering, Dankook University, Cheonan 31116, Korea; nhlee0609@dankook.ac.kr (N.-H.L.); ysm6610@dankook.ac.kr (S.Y.); a.taghizadeh@dankook.ac.kr (A.T.); m.taghizadeh@dankook.ac.kr (M.T.); 2Department of Nanobiomedical Science and BK21 NBM Global Research Center for Regenerative Medicine, Dankook University, Cheonan 31116, Korea; 3Department of Chemistry, Dankook University, Cheonan 31116, Korea; 4Department of Regenerative Dental Medicine, College of Dentistry, Dankook University, Cheonan 31116, Korea; 5Mechanobiology Dental Medicine Research Center, Dankook University, Cheonan 31116, Korea

**Keywords:** cell membrane, nanoparticle, drug delivery, cell membrane engineering

## Abstract

Cell membrane cloaking technique is bioinspired nanotechnology that takes advantage of naturally derived design cues for surface modification of nanoparticles. Unlike modification with synthetic materials, cell membranes can replicate complex physicochemical properties and biomimetic functions of the parent cell source. This technique indeed has the potential to greatly augment existing nanotherapeutic platforms. Here, we provide a comprehensive overview of engineered cell membrane-based nanotherapeutics for targeted drug delivery and biomedical applications and discuss the challenges and opportunities of cell membrane cloaking techniques for clinical translation.

## 1. Introduction

Cell membrane-cloaked nanotherapeutics integrated with the biomimetic features of cell membranes with multifunctional nanoparticles emerged as a future-oriented platform for targeted drug delivery. They can inherently reproduce the biological properties of the source cells and achieve a wide range of functions, such as prolonged circulation, immune escape, and disease-relevant targeting ability [1,2,3,4,5,6,7]. The traditional gold standard technique for surface modification of nanoparticles is poly (ethylene glycol) (PEG) coating. PEGylated nanoparticles show improved stability, pharmacokinetics, and immunogenicity of the nanoparticles in vivo as compared to that of uncoated nanoparticles [8]. However, recent studies reveal that PEG-coated nanoparticles mediate complement activation and recognition by PEG-specific IgM antibodies, leading to hypersensitivity in vivo and accelerated blood clearance by the liver [9]. The cell membrane cloaking helps to overcome such drawbacks associated with traditional surface modification techniques and facilitates favorable accumulation in the target tissue [10]. Moreover, inherent properties of cell membranes allow cell membrane-coated nanoparticles to have distinct biointerfacing, such as cell-specific interactions and in vivo pharmacokinetics. For example, red blood cell membrane-coated nanoparticles have superior prolonged circulation in vivo, while macrophage membrane coating of nanoparticles facilitates accumulation in inflammatory sites as compared to that of the RBC membrane coating [11]. Recently, to expand the use of cell membrane coating techniques in clinical application, cell membrane functionalities are often modified via physical, chemical, and genetic engineering [12,13,14,15,16]. Furthermore, versatile nanotherapeutic platforms, such as mesoporous silica nanoparticles, magnetic nanoparticles, and metal-organic framework, were combined with cell membrane coating technology for finely controlled drug delivery and combinatory therapy, leading to promising clinical implications in molecular imaging, regenerative medicine, and cancer immunotherapy [1,3,4,6,17,18,19,20,21,22,23,24]. Here, we focus on the recent progress in cell membrane-cloaked nanotherapeutics for biomedical applications (Figure 1) and discuss perspectives for the future clinical translation of cell membrane-cloaked biomaterials.

## 2. Cell Membrane Cloaking Technology

Cell membrane cloaking is a platform technology that makes nanoparticle surfaces directly replicate the complex biointerfacing of the source cells. The functionalities of the cell membrane-cloaked nanoparticles are largely influenced by the inherent properties of the source cells. In the following section, we introduce source cell types for cell membranes and their unique features helpful for targeted drug delivery, phototherapy, immune modulation, and theragnostic [23,24,25,26,27,28,29].

### 2.1. Natural Cell Membranes and Their Unique Features

Cell membrane source is carefully selected for surface modification of nanoparticles depending on target diseases. Inherent properties of cell membranes allow cell membrane-coated nanoparticles to have unique biointerfacing characteristics such as cell-specific interactions and in vivo kinetics. In this section, we introduce natural cell membranes and their inherent features that are useful for targeted drug delivery (Table 1). 

#### 2.1.1. Red Blood Cell Membrane

Red blood cell (RBC) or erythrocyte is the major source of the blood and plays a role in oxygen transportation and detoxification. RBCs are disciform with a hollow center, allowing resistance to osmotic pressure or external shock and flexibility to pass through the capillary. RBC membranes were extensively used for modifying nanoparticles due to their ability to avoid the immune system and achieve long-term circulation [2,25,27,28,29,30,31]. For example, Hu et al. [32] reported that RBC membrane-coated nanoparticles had significantly longer circulation time as compared to that of bare nanoparticles or RBC membrane vesicles. Without RBC membrane coating, bare nanoparticles rapidly aggregated in serum within a few minutes after intravenous injection. RBC membrane vesicles were also removed from the bloodstream in less than 30 min [33]. This result suggests that introducing a solid core into RBC membrane vesicles would contribute to the higher structural rigidity and better particle stability, leading to the prolonged circulation time of RBC membrane-coated nanoparticles. Furthermore, the RBC membrane coating is confirmed to be superior in retarding in vivo clearance as compared to that of the conventional PEG stealth coating due to the immunosuppressive proteins of RBC membranes that inhibit macrophage [34].

Notably, RBC membrane-coated nanoparticles were used for detoxification [35]. Nanoparticles coated with RBC membranes can act as decoys to adsorb and neutralize varieties of bacterial pore-forming toxins (PFTs), protecting healthy RBCs from being attacked [36,37]. Interestingly, although RBC membrane vesicles alone can adsorb toxins, they fail to reduce the hemolytic activity of toxins due to their fusion with healthy RBCs. On the other hand, RBC membrane-coated nanoparticles can not only arrest toxins, but also prevent fusions with other RBCs, highlighting the role of the core in the formulation. Recently, inspired by the unique toxin neutralization property of RBC membranes, a stimuli-responsive drug release strategy was combined into one nanotherapeutic platform for detoxification therapy [35,38]. The RBC membrane shell adsorbs and neutralizes PFTs secreted by bacteria, while the core releases antimicrobial drugs in a controlled manner. Details of RBC membrane-coated nanotherapeutics are described in Section 4. 

#### 2.1.2. Platelet Membrane

Platelet acts as circulating sentinels for vascular damage and invasive microorganisms. Its envelope is composed of glycoproteins that take charge of attachment and agglomeration of platelets, while its inside is composed of granules containing various substances and coagulants involved in its function, such as blood clotting. Due to the inherent platelet properties such as immunocompatibility, specific binding affinity to injured vasculature, and pathogen adhesion, platelet membranes were used to functionalize nanoparticles, leveraging their natural binding ability to design a targeted drug delivery system for bacteria-, cancer-, and blood vessel-related disorders [39,40]. Hu et al. [40] reported that platelet membrane-coated nanoparticles selectively adhered to damaged vasculatures as well as enhanced binding to platelet-adhering pathogens. Furthermore, compared to that of uncoated nanoparticles, the platelet membrane-coated nanoparticles reduced cellular uptake by macrophages and lack nanoparticle-induced complement activation in autologous human plasma. Recently, Ma et al. [41] reported that the platelet membrane coating of nanoparticles facilitated the specific accumulation of the nanoparticles in early-stage plaque sites by specific binding to plaque-infiltrated macrophages, rather than normal vessels. Furthermore, platelet membrane-coated nanoparticles not only significantly prolonged the blood retention time but also had minimal influence on liver and kidney functions even at higher doses and with continuous administration. 

#### 2.1.3. Macrophage Membrane

Macrophages (MΦs) are important cells of innate immunity and are known for their phagocytic activity, capability for antigen presentation, and flexible phenotypes. Due to their inherent affinity to inflammation sites, MΦ membrane-coated nanoparticles are likely to be accumulated in chronic inflammatory sites such as cancer, gout, and atherosclerosis [11,42]. Unlike RBC membrane-coated nanoparticles, MΦ membrane-coated nanoparticles not only facilitate accumulation in inflammatory sites but also efficiently neutralize inflammatory cytokines, indicating MΦ membrane coating strategy is more suitable for targeting inflammatory sites rather than the RBC membrane coating strategy [11]. So far, most of the studies used macrophage membranes derived from allogeneic cells or cell lines and often resulted in unwanted biological effects for targeting tumors, such as immune incompatibility and immunogenicity issues, limiting their clinical application [43,44]. Recently, Chen et al. [45] reported that tumor-associated macrophages (TAM) membranes derived from primary tumors had unique antigen-homing affinity capacity and immune compatibility. Macrophage colony-stimulating factor 1 (CSF1) secreted by cancer cells binds to its receptor (CSF1R) on the macrophage membrane, and in turn, activates the downstream signaling pathway responsible for the polarization of TAMs to the immunosuppressive phenotype. TAM membrane-coated nanoparticles could scavenge CSF1 secreted by cancer cells due to the high CSF1R content on the TAM membranes, blocking the interaction between TAMs and cancer cells and thereby enhancing the efficacy of the nanoparticles for cancer immunotherapy. 

#### 2.1.4. Cancer Cell Membrane

Cancer cell membranes have strong tumor-targeting potential due to the retained membrane structure and antigens of cancer cells. Cancer cells overexpress growth factor receptors (e.g., EGFR and IGF-1) and adherent proteins (e.g., integrins, CD44, and CD24) involved in tumor progression and invasion on their cell membranes. This contributes to cancer cells membrane-specific characteristics such as homotypic targeting, complex antigenic profile, and low intrinsic immunogenicity [11,46]. Depending on the type of cancer being targeted, the type of cancer cell membrane used for nanoparticle coating varies. In many cases, the same type of cancer membrane is used as target cancer due to the homologous targeting abilities, displaying superiority in cancer imaging and targeted therapy [5,11,47]. Despite their excellent homologous targeting abilities, cancer cell membranes often fail to induce the maturation of antigen-presenting cells [48]. For effective chemo-immunotherapy, cancer cell membranes were modified. Wu et al. [49] reported that cancer cell membrane-coated nanoparticles induced an enhanced immune response by incorporating bacteria-derived proteins. The bacteria-derived protein decorated acted as natural adjuvants and activated immune response by increasing T cell proliferation and cytokine secretion. Therefore, nanoparticles coated with cancer cell membranes decorating bacteria-derived proteins exhibited higher antitumor growth efficacy compared with that of ones coated with cancer cell membranes alone. Likewise, there were various attempts to engineer cancer cell membranes for improved cancer therapy [46]. Details are described in Section 2.2.

#### 2.1.5. Cell Organelle Membrane

Membranes of cell organelles such as mitochondria, nucleus, endoplasmic reticulum, Golgi apparatus, and lysosomes share the same fundamental structure as the plasma membranes [50,51]. The cell organelle membrane coating of drug carriers can be beneficial for improved therapeutic efficiency. For example, coating nanoparticles with nuclear, mitochondrial, or lysosomal membranes can bypass anticancer drug resistance of the downstream pathways in cancer treatment. Nuclear membrane-coated nanoparticles can improve transfection efficiency in gene therapy. Gong et al. [52] recently reported that mitochondrial membrane-coated nanoparticles could selectively bind mitochondrial membrane ligands and neutralize toxins. Recently, cell organelle membranes started to be used for nanoparticle surface modification. Considering the great potential of cell organelle membranes, future studies will open new therapeutic opportunities.

#### 2.1.6. Bacterial Membrane

Bacterial membrane-coated nanoparticles recently emerged as a unique antivirulence approach against infectious diseases [53,54]. Bacterial membranes contain a large number of immunogenic antigens with intrinsic adjuvant properties [55], delivering effector molecules critical for pathogen dissemination such as pathogen-associated molecular patterns (PAMP) and other virulence factors to host cells [56,57,58]. Although bacterial membrane itself is an appealing vaccination material, its vaccination effect remains elusive due to its low structural stability and poor size homogeneity. The fragile vesicle structure of the bacterial membrane is not favorable for keeping integrity, leading to poor stimulation of antigen-presenting cells. Bacterial membrane-coated nanoparticles were developed in many forms of ‘nanovaccines’. Attaching bacterial membranes to nanoparticles generates structurally stable vaccines with uniform sizes and improves the immune efficacy of bacterial membranes for vaccine development [59,60]. For example, Gao et al. [60] demonstrated that bacterial membrane-coated gold nanoparticles induced rapid activation and maturation of dendritic cells in the lymph nodes of the vaccinated mice and generated antibody responses that are durable and of high avidity than those elicited by a bacterial membrane only, resulting in a strong Th1 and Th17 biased cell responses against the source bacteria.

Noteworthily, bacterial membrane-coated nanoparticles were recently used as a ‘nanodecoy’ for antibacterial adhesion therapies. The bacterium-mimicking nanoparticles compete with the source bacteria for binding the host [61], and such weakening bacterial adhesion facilitates the host immune system to eliminate the pathogen [62]. Thus, the antiadhesion nanomedicine platform is less likely to propagate resistance when compared with that of conventional antibiotics. Zhang et al. [63] reported that Helicobacter pylori membrane-coated nanoparticles reduced H. pylori adhesion to gastric epithelial cells and such antiadhesion efficacy was dependent on the dose of the nanoparticles. Adopting multifunctional nanoparticle cores to the antivirulence approach in combination with bacterial membrane coating will offer exciting opportunities and provide numerous implications toward effective and safe antibacterial therapies.

### 2.2. Cell Membrane Engineering Technique

There are three major engineering strategies categorized as physical, chemical, and genetic. The physical engineering of cell membranes relies on lipid fusion, such as lipid-anchoring methods where insert lipid-conjugated molecules, functional groups, or therapeutics to lipid bilayers of cell membranes. Additionally, cell membrane vesicles are fused with liposomes, extracellular vesicles, or cell membrane vesicles from different cell sources. The chemical engineering strategies are based on covalent conjugation of functional groups, moieties, or therapeutics to the residues of membrane-associated proteins and polysaccharides via various chemical reactions such as EDC/NHS coupling reaction [64,65,66], thiol-ene reaction [67], biotin-avidin recognition [68], and bio-orthogonal click chemistry [69]. The genetic engineering of the cell membrane enables much more complex multicomponent modification. The following section describes cell membrane engineering techniques which are the most frequently used in this field. 

#### 2.2.1. Hybridization

The hybrid membrane can be simply prepared by fusing cell membranes from different cell sources with sonication and mechanical extrusion, and thus can perform increasingly complex tasks within biologically relevant contexts [2,70,71]. Thus, a hybrid membrane coating strategy could be a facile and effective way to enhance nanoparticle functionality [72]. For example, RBC-cancer cell hybrid membranes exhibited both long circulation and homotypic tumor-targeting properties after being coated onto nanoparticles [2]. Retinal endotheliocyte-RBC (REC-RBC) hybrid membranes obtained homotypic targeting ability, binding ability to the vascular endothelial growth factors, and immune evasion [70]. Therefore, REC-RBC membrane-coated nanoparticles drastically accumulated in choroidal neovascularization regions as compared to REC or RBC membrane-coated nanoparticles. In another study, platelet-white blood cell (PLT-WBC) hybrid membrane-coated nanoparticles were demonstrated to inherit enhanced cancer cell binding ability and reduced homologous WBC interaction from PLT and WBC, respectively [71]. Targeted drug delivery and therapeutic effects of hybrid membrane-based nanotherapeutics in detail are described in Section 3.

#### 2.2.2. Lipid-Anchoring Method

Lipid-conjugated ligands or therapeutic molecules can be anchored to cell membranes through lipid–lipid interaction, called a lipid-anchoring method [73]. This method can easily modify cell membranes with various molecules as compared with chemical engineering methods. For example, Gao et al. [16] modified RBC membranes with lipid-conjugated T807 (DSPE-PEG3400-T807) and triphenylphosphine (DSPE-PEG2000-TPP) by gently incubating under mild conditions. Nanoparticles coated with this membrane (T807/TPP-RBC-NPs) increased accumulation in neurons after crossing the brain-blood barrier due to the navigation effects of T807, and further localized to the mitochondria due to the aid of TPP, a mitochondrion targeting ligand, as compared to that of naïve RBC membrane-coated nanoparticles. Thus, curcumin, an antioxidant-loaded T807/TPP-RBC-NPs, efficiently relieved Alzheimer’s disease symptoms by mitigating mitochondrial oxidative stress and suppressing neuronal death. In another study, Yang et al. [15] modified cancer cell membrane with mannose moiety by a lipid-anchoring method to enhance antitumor immune responses. PLGA nanoparticles were loaded with R837, a toll-like receptor agonist as an adjuvant, and then coated with the cancer cell membrane modified by mannose as an immune recognition moiety, acting as a cancer vaccine. The released R837 from the nanoparticles promoted the immunogenicity of the nanoparticles, and surface-modified mannose facilitated the cellular uptake of the nanoparticles by antigen-presenting cells (APCs) via specific binding between mannose and its receptors on APCs; it further enhanced the lymph node retention of the nanoparticles in vivo. 

#### 2.2.3. Genetic Engineering

Despite the ease of lipid-anchoring methods, the method often challenges when it comes to displaying large transmembrane protein receptors or ligands at a high density due to hindrance effects [73]. Genetic engineering can directly introduce desired proteins or peptides on cell membranes through transfection or transduction via nonviral or viral vectors, respectively, resulting in selective transgene protein expression at a relatively high surface density. For example, Ma et al. [12] engineered neural stem cells (NSCs) to overexpress CXCR4 through lentiviral transduction [12]. In comparison to naïve NSC membrane-coated PLGA nanoparticles, CXCR4-overexpressed NSC membranes-coated ones selectively accumulated in the ischemic microenvironment through chemotactic interaction with SDF-1, a ligand of CXCR4, enriched in the ischemic region. This led to improved stroke-targeting delivery of cargo therapeutics, glyburide, and promoted stroke repair and recovery. In another study, Park et al. [14] engineered wild-type cells to express very late antigen-4 (VLA-4), which is composed of integrins α4 and β1. The resulting VLA-4-expressing cell membrane-coated nanoparticles can efficiently target VCAM-1 on inflamed lung endothelial cells for enhanced drug delivery. Jiang et al. [13] engineered a murine melanoma cell line to overexpress two sets of genes, a cytosolic form of ovalbumin (OVA) as a model antigen, and a costimulatory marker CD80, which engages the CD28 receptor found on T cells [13]. Nanoparticles coated with this cell membrane presented two signals together, promoting activation of the cognate T cells via an antigen peptide-MHC complex and CD80. Such tumor antigen-specific immune responses triggered by the nanoparticles coated with OVA and CD80-overexpressed cancer cell membranes significantly inhibited tumor growth in vivo. These complex and multicomponent membrane-bound ligand modifications would be infeasible to incorporate by traditional synthetic strategies, such as lipid-anchoring methods. Cell membrane functionalities can be finely crafted via genetic engineering and further expand the wide-ranging utility of cell membrane coating technology in the near future. 

## 3. Various Nanoparticle Cores for Controlled Drug Delivery and Combinatory Therapy

Combined with various nanoparticle cores, the cell membrane-cloaked technique can leverage the clinical potentials of nanotherapeutics. Ideal nanoparticle cores not only act as a drug carrier but also possess multifunctionalities for combinatory therapy. In the following section, we describe recent progress in the development of nanotherapeutic platforms combined with the cell membrane cloaking technique (Table 2).

### 3.1. Nanoparticles Incorporating Drugs

#### 3.1.1. Polymeric Nanoparticle

The most popular inner core for cell membrane cloaking is polymeric nanoparticles composed of poly (lactic-co-glycolic acid) (PLGA), PLGA is a biodegradable and biocompatible copolymer that is approved by FDA. PLGA is hydrolyzed to lactic acid and glycolic acid monomers in the body [87], and the byproducts are nontoxic and can be metabolized by the body [88]. Furthermore, its degradability and degradation profile can be controlled by adjusting the ratio between lactic acid and glycolic acid [88]. A double emulsification method is often used for encapsulating a wide range of drugs with varying solubilities in PLGA nanoparticles [89]. Drugs are dissolved in an aqueous solvent and are subsequently added to PLGA dissolved in a volatile organic solvent in a dropwise manner, forming primary water in oil (W/O) emulsion. The mixture is then added to another aqueous phase containing a stabilizer (e.g., PVA) and mixed by stirring, forming a water-in-oil-in-water (W/O/W) emulsion. The size of PLGA nanoparticles can be adjusted by the drug-to-PLGA ratio, the type of organic solvents, the stabilizer concentration, stirring speed, and so on, which eventually affects the drug release profiles of PLGA nanoparticles [87]. Many researchers developed drug delivery systems based on cell membrane-coated PLGA nanoparticles [2,12,14,32,71]. Interestingly, they reported that the cell membrane coating of PLGA nanoparticles affected the kinetics of drug release. For example, Bose et al. [81] reported that stem cell membrane coating of PLGA nanoparticles encapsulating VEGF slowed down the VEGF release from 35% to 15% within a day, and from 70%–80% to 55% by day 30, respectively, as compared to that of bare ones without a cell membrane coating. Similarly, Wang et al. [83] reported that macrophage cell membrane-coated PLGA nanoparticles exhibited a slightly slower drug release profile as compared to that of the nanoparticles without a cell membrane coating. The surface of the cell membrane-coated nanoparticles can act as a diffusion barrier, and thus the cell membrane-coated nanotherapeutics can achieve steady and long-term drug delivery.

Gelatin is also widely used as a nanoparticle material due to its gel-forming ability, biodegradability, and biocompatibility [90]. Its pH-dependent gelation allows easy drug loading of gelatin nanoparticles [91]. The drug loading efficiency of gelatin nanoparticles depends on the isoelectric point of gelatin and the electrostatic properties of the drug. Furthermore, the drug release kinetics of gelatin nanoparticles are largely influenced by the gelatin degradation which can be tuned by the molecular weight and crosslinking extent of nanoparticles [90,91]. Rao et al. [78] prepared tumor cell membrane-coated gelatin nanoparticles for tumor microenvironment-triggered drug release. The gelatin core was degraded by matrix metalloproteinases (MMPs) at tumor tissues, accelerating the drug release from the nanoparticles. Furthermore, the nanoparticles that entered cells were degraded in acidic endosomes and further facilitated the drug release. They noted that the cell membrane coating of gelatin nanoparticles did not negatively impact drug release from the nanoparticles.

#### 3.1.2. Mesoporous Silica Nanoparticle

Mesoporous silica nanoparticles (MSNs) are excellent drug carriers due to their extremely large surface area and biodegradability [92]. The large surface area of MSNs allows nonspecific adsorption of drugs, showing superior drug loading to other carriers [93]. In addition, MSNs can be tailed to exhibit spatiotemporal control of drug release by modifying the surfaces with stimuli-responsive polymers, gatekeepers, and targeting ligands [18,19,94,95,96,97]. For example, Zhang et al. [18] reported that dendritic MSNs could achieve tumor site-specific drug release due to their robust degradation under acidic and reduction conditions such as tumor microenvironment. Furthermore, drugs were anchored on the surface of dendritic MSNs via a disulfide bond, accelerating the drug release in the glutathione-rich cytoplasm. Shao et al. [95] introduced diselenide bond-containing organosilica moieties into the MSN framework to achieve oxidative and redox dual-responsive degradation properties. The diselenide-bridged MSNs released encapsulated RNase A in response to the oxidative/redox tumor microenvironment, which led to reduced tumor volumes and tumor weights compared with the free RNase A-treated group. Cancer cell membrane coating of MSNs helps with escaping the immune clearance, prolonging circulation time, enhancing tumor accumulation, and improving intracellular uptake of the nanotherapeutics [18,19,95]. Nie et al. [19] reported that cancer cell membrane-coated MSNs were directly internalized to cancer cells by membrane fusion and the MSNs were subsequently released into the cytosol, indicating a similar internalization pathway as an enveloped virus. Of note, the core MSNs were coated with PEG-liposomes before the cancer cell membrane coating, which allowed the nanoparticle to transport its loaded anticancer drugs to the nucleus by intracellular trafficking throughout the cytoskeletal filament network, whereas the noncoated MSNs diffused only into the peripheral zone of the cytoplasm. Taken together, surface modification of MSNs should be carefully tailored to achieve efficient drug delivery to target cells as well as subcellular organelles.

#### 3.1.3. Drug Cluster

Polymeric or inorganic particles are often used as a drug carrier. However, in most cases, the proportion of drug payloads in the formulations is very small, typically <10%, as compared to the quantities of carrier materials. It also can cause material-associated cytotoxicity. High drug-carrying capacity and loading yield are important to further achieve therapeutically effective doses at the targeted sites. Thus, there was an attempt to cluster drugs and form drug aggregates as nanoparticles and improved the drug loading content [98,99,100]. Pei et al. [77] prepared a drug aggregates-based inner core constructed by methoxypoly(ethylene glycol)-block-poly(D, L-lactide) (PEG-b-PDLLA), reactive oxygen species (ROS)-responsive paclitaxel (PTX) dimer and photosensitizer, 5,10,15,20-tetraphenylchlorin (TPC). The drug loading content of PTX dimer in the formulation was 85 wt%. Upon light irradiation, the TPC generated ROS, and it triggered the cleavage of the PTX dimer, leading to PTX release. Similarly, Chai et al. [80] prepared a docetaxel nanocrystal (NC(DTX)) as a core and obtained extremely high drug loading yields. Due to the intrinsically unstable and insoluble nature of DTX, NC(DTX) alone caused severe toxicities [101]. In contrast, RBC membrane-coated NC(DTX) significantly reduced systemic toxicity and achieved long-term stability in serum.

### 3.2. Multifunctional Nanoparticle Cores

#### 3.2.1. Gold Nanoparticle

Gold nanoparticles possess unique electrical, physical, and optical properties according to their shapes (e.g., nanospheres, nanorods, and nanocubes) and sizes (2–150 nm). Their adjustable light absorption and scattering abilities in both visible and infrared regions make them relevant to be used for imaging, sensing, and photodynamic therapy (PDT) in cancer research [60,79,102]. For example, Hu et al. [102] prepared a cell membrane-functionalized nanoprobe using gold nanoparticles for imaging apoptosis. FITC-labeled caspase-3 substrate peptide was anchored onto the surface of gold nanoparticles and subsequently coated with the cancer cell membrane. Due to the proximity of FITC to gold nanoparticles, the fluorescence of FITC was quenched and the nanoprobe signal is off. When the nanoprobe was internalized to cells, the caspase-3 substrate peptide was cleaved by active caspase-3 and the fluorescence signal of FITC is back on. In this regard, the cell membrane-coated nanoprobe can detect caspase-3 activity, an indicator of cell apoptosis, and image cancer cells. In another study, Sun et al. [79] reported that cancer cell membrane-coated gold nanorods exhibited robust photothermal transferability and radiosensitizing ability due to its gold nanorod core. Furthermore, the cancer cell membrane coating of the gold nanorods improved specific tumor targeting and perinuclear accumulation. Under the near-infrared (NIR) light and X-ray irradiation, the cancer cell membrane-coated gold nanorods significantly suppressed tumor growth by a temperature increase and ROS generation. 

#### 3.2.2. Magnetic Nanoparticle

Magnetic nanoparticles are widely used as contrast agents for magnetic resonance imaging (MRI) due to their superparamagnetic properties [24,103]. Furthermore, they possess broad photoadsorption in the NIR range and thus are often used as a photosensitizer for photothermal therapy (PTT) [3,20,21,104]. For example, Bu et al. [20] used FDA-approved iron oxide nanoparticles, Fe_3_O_4_, as a core and coated them with platelet-cancer stem cell hybrid membrane for enhanced photothermal therapy of head and neck squamous cell carcinoma (HNSCC). The platelet membrane and cancer stem cell membrane provided immune evading and homotypic targeting capabilities, respectively. Thus, the hybrid membrane-coated magnetic nanoparticles induced higher cancer cell uptake and lower macrophage uptake as compared to that of PEGylated ones. Due to the high concentration of Fe accumulated in tumor sites, the local temperature of tumors treated with the hybrid membrane-coated magnetic nanoparticles drastically raised to 58.3°C within 5 min upon NIR laser exposure, leading to excellent HNSCC tumor growth inhibition. More recently, it was reported that ferumoxytol, a FDA-approved Fe_3_O_4_ nanoparticle, can transform M2 macrophages to M1 macrophages by the Fenton reaction which generates highly toxic hydroxyl radicals (OH^•^) by a reaction between Iron (II) (Fe^2+^) and hydrogen peroxide (H_2_O_2_) [105]. Therefore, the increased number of M1 macrophages could improve antitumor immune response in the tumor microenvironment. Yu et al. [21] took advantage of the capabilities of Fe_3_O_4_ magnetic nanoparticles such as macrophage polarization and PTT-induced immunologic cell death for enhanced antitumor immune response. By coating the nanoparticles with myeloid-derived suppressor cells (MDSCs), which are a major regulator of immune responses in cancer, MDSC membrane-coated nanoparticles allowed efficient immune escape and targeting to the tumor site as compared to that of RBC-coated ones. Therefore, the MDSC membrane-coated nanoparticles enabled high-contrast tumor imaging with MRI for diagnosis. Furthermore, this magnetic nanoparticle-based system demonstrated excellent performance in PTT-induced tumor growth inhibition by IR laser irradiation. 

Hybrid magnetic nanoparticles were used to achieve a synergistic effect on drug delivery. For example, Xuan et al. [3] prepared a magnetic MSN by attaching magnetic colloids onto the surface of MSNs, and photosensitizer, hypocrellin B (HB), was subsequently encapsulated to the magnetic MSN. The magnetic MSN exhibited high HB loading due to the large surface area of MSNs. More importantly, by coating the HB-loaded magnetic MSN with RBC membrane, the nanoparticles held 90% of HB loaded inside of the nanoparticles, while uncoated ones only kept 20% of HB. In combination with photosensitizer delivery, the external magnetic field intervention and light irradiation led to a robust nanoparticle accumulation in tumor tissues and improved the performance of photodynamic therapy (PDT), which led to completely suppressed tumor growth. Similarly, Ding et al. [104] used Fe_3_O_4_ nanoparticles coated with a SiO_2_ layer (6.5 nm) (Fe_3_O_4_@SiO_2_) as a core of their platform for ultrasensitive isolation and detection of circulating tumor cells (CTCs). The surface of Fe_3_O_4_@SiO_2_ was coated with white blood cell-tumor cell hybrid membranes and was further grafted with multivalent aptamer-Ag_2_S nanodots, a NIR fluorescence biosensor, achieving both magnetic and NIR fluorescence properties. Fe_3_O_4_@SiO_2_ exhibited a fast-magnetic response as Fe_3_O_4_ nanoparticles. Furthermore, the dual coating Fe_3_O_4_ with the cell membrane and SiO_2_ shell efficiently blocked the energy transfer between Ag_2_S NDs and magnetic nanoparticles, increasing the sensitivity of detection and bioimaging. Moreover, captured CTCs by the nanoparticle were easily separated with magnetic isolation. 

#### 3.2.3. Metal–Organic Framework

Recently, metal–organic frameworks (MOFs) attracted much attention in biomedical fields due to their potential applications for drug delivery, molecular imaging, and biological sensing. MOF is classified as a hybrid porous nanomaterial that is composed of metal ions or metal clusters combined with organic ligands. The size, structure, surface functionality, and physical properties of MOF can be tuned depending on the type of organic components [4,6,22]. MOFs can serve as an excellent drug carrier due to their large surface area, tailored pore size, pre-designed morphology, and controlled degradability. MOFs can embed drugs and biomolecules in their tight cavities with high loading efficiency [1,4,6,22], and such confinement encapsulation could significantly reduce the structural deformation of drugs and biomolecules and prevent their leaching [17]. Therefore, MOFs were used as a carrier for various drugs such as an enzyme, small molecule [22], genome editing machinery [1], and siRNA [17]. For example, Zhang et al. [22] delivered glucose oxidase (GOx) and tirapazamine (TPZ) using zeolitic imidazolate framework-8 (ZIF-8) to tumor sites. The ZIF-8 was confirmed to reserve the enzymatic activity of GOx and released GOx under acidic conditions, but not under physiological conditions. RBC membrane cloaking of the ZIF-8 significantly improved the retention time and immune escape led to efficient tumor accumulation. The GOx released from the ZIF-8 starved tumors by exhausting endogenous glucose and oxygen in tumor cells. The GOx-induced starvation subsequently initiated the activation of TPZ and further inhibited tumor growth. Similarly, Alyami et al. [1] prepared cancer cell membrane-coated ZIF-8 for targeted and cell-specific delivery of CRISPR/Cas9. Despite the huge size of CRISPR/Cas9, it was successfully encapsulated in ZIF-8. Due to the cancer cell membrane coating, the nanotherapeutics showed a higher uptake by cancer cells than by healthy cells. In another study, it also was confirmed that MOFs not only increased the drug loading with high loading yields but also accelerated drug release in cytosols due to their pH-sensitive degradability [17]. 

MOFs not only can serve as a carrier but also play as a photosensitizer for photodynamic therapy (PDT). Min et al. [6] loaded apatinib in ZIFs, wrapped them in MnO_2_ and subsequently coated them with cancer cell membranes. At tumor sites, the MnO_2_ shell consumed the excessive GSH, sensitizing tumor tissues for the subsequent PDT effect of ZIFs. In addition, the reaction product Mn^2+^ could be used as an MRI contrast agent for in vivo tumor imaging. As the MnO_2_ shell was degraded, apatinib was released from the ZIF and neutralized the PDT-induced revascularization, preventing tumor progress. Similarly, Gao et al. [4] took advantage of the photosensitization and gas-adsorption capability of MOFs to develop oxygen-evolving PDT nanoplatforms. Zirconium (IV)-based MOF (UiO-66) was used as a vehicle for O_2_ storing, then conjugated with indocyanine green (ICG) and further coated with RBC membranes. Upon 808 nm laser irradiation, the initial singlet oxygen (^1^O_2_) generated by ICG decomposed RBC membranes, and the photothermal properties of ICG facilitated the burst release of O_2_ from UiO-66. The generated O_2_ significantly improved the PDT effects on hypoxic tumors. Taking advantage of multifunctional nanoparticles, multifunctional and versatile nanotherapeutic platforms can be developed, and the drug delivery efficiency can be synergistically improved in combination with cell membrane cloaking techniques.

## 4. Cell Membrane-Cloaked Nanotherapeutics for Various Diseases

### 4.1. Cancer

The advent of cell membrane cloaking technique in cancer nanotechnology bestows nanoparticles with extended blood circulation period, enhanced immune evasion, localized drug delivery, improved tumor penetration, and accumulation [43,106]. Nanoparticles with a high capacity of anti-cancer drug loading and photodynamic properties were decorated with cell membranes isolated from various cell types, including cancer cells, fibroblasts, platelets, macrophages, RBCs, white blood cells, dendritic cells, and natural killer (NK) cells [7,23,24,27,29,31,107,108]. Cell membranes obtained from cancer cells and immune cells were extensively used especially for cancer immunotherapy due to their inherent immune evasion and homologous tumor-targeting ability [49,109]. Furthermore, as discussed in Section 1, engineered cell membranes were used to increase the targeting efficiency and therapeutic effects of nanotherapeutics, especially for cancer immunotherapy. To boost antitumor immunity, cell membranes are often engineered to constitutively express stimulatory cues (e.g., ovalbumin, CD80 and SIRPa) for antigen-presenting cells (APCs) and tumor-associated macrophages (TAMs). Nanoparticles coated with membranes of cancer cells, which naturally present their own antigens via MHC-I, engineered to express both CD80, a costimulatory signal, and ovalbumin (OVA), an antigen, can act as an artificial APCs and promote the activation of the cognate T cells as compared with that of ones coated with wild-type cancer cell membranes [13]. As compared with that of wild-type cancer cell membrane-coated nanoparticles or single knock-in CD80- or OVA-nanoparticles, the double knock-in CD80/OVA-nanoparticles significantly induced cytokine secretion (i.e., IL-2 and IFN-γ) and increased the cytotoxicity of CD8^+^ T cells against target cancer cells, leading to inhibition of tumor growth in vivo. Recently, Chen et al. suggested an interesting strategy called the “cascade cell membrane coating” technique for manipulating T cell immunity [110]. Dendritic cells were prepulsed with cancer cell membrane-coated nanoparticles, and thereby the DC membrane could present an array of cancer cell membrane antigen epitopes. The cascade cell membrane retains the membrane composition and intrinsic function of both cancer cells and DC membranes, bypassing the need for regular antigen processing. Therefore, nanoparticles coated with this DC membrane elicited strong antigen-specific T cell responses and tumor-specific immunity in various tumor models. Noteworthily, combination therapy of this nanotherapeutic with anti-programmed cell death-1 (αPD-1) antibodies, a clinical immune checkpoint inhibitor, synergistically inhibited tumor growth and improved survival rates while αPD-1 alone failed to do so.

In the tumor microenvironment, cancer cells trick macrophages by expressing CD47, a “don’t eat me” signal, on their cellular surface and protect them from phagocytosis via binding to signal regulatory protein alpha (SIRPα) receptor on macrophages. In other words, blockade of CD47-SIRPα signaling can promote macrophages to directly “eat” cancer cells. Rao et al. [111] demonstrated that magnetic nanoparticles coated with cell membranes genetically overexpressing SIRPα enhanced affinity to CD47 and blocked CD47-SIRPα signaling pathway. The magnetic core improved the accumulation in the tumor site under magnetic navigation, reduced the off-target immune overactivation. Interestingly, the in vivo results revealed that the treatment of SIRPα-overexpressed cell membrane-coated magnetic nanoparticles significantly inhibited the tumor growth, while single or coadministration of magnetic nanoparticles and the cell membrane showed limited antitumor effects, indicating that the enhanced tumor accumulation of the nanoparticles via magnetic navigation is largely attributed to the improved antitumor effects. This also led to increased cellular uptake of the nanoparticles by TAMs, resulting in the repolarization of TAMs from protumorigenic M2 phenotype to antitumor M1 one, and further contributed to regaining the antitumor immunity of CD47 blockades. Therefore, in melanoma and breast cancer models, this nanotherapeutic significantly prolongs overall mouse survival by inhibiting primary tumor growth and distant tumor metastasis. Taken together, recent progress demonstrates that the combination of cell membrane cloaking technology and multifunctional nanoparticle-based drug delivery platforms offer a safe and robust strategy in activating host immune responses for cancer immunotherapy.

Among various nanoparticles, MOFs recently drew much attention as a novel nanotherapeutic platform in cancer therapy. Unlike conventional nanoparticles applied in cancer treatment, distinctive properties of MOFs such as high drug loading capacity, stimuli responsiveness, and easy manipulation of compositional and structural properties enable the combination therapy of chemotherapy, PDT, PTT, enzymatic reaction, and immunotherapy [112]. For instance, in an interesting study by Sun et al. [113], a hollow porphyrinic zirconium-based MOF (PCN-222) with a high PDT property was developed and loaded with a high concentration of indocyanine green (ICG) and doxorubicin (DOX) for PTT and chemotherapy, respectively. By coating MOF with cell membranes of a 4T1 breast cancer cell line, the tumor uptake of the MOF was enhanced, as compared to that of uncoated MOFs as injected intravenously to 4T1-xenografted mice. Furthermore, the drug release of the MOF was facilitated upon NIR (808 nm) irradiation or/and under acidic conditions such as tumor microenvironment. More importantly, improved singlet oxygen generation from the hollow MOF under 660 nm laser irradiation significantly decreased the viability of 4T1 cells in vitro and also inhibited in vivo tumor growth, suggesting superior combinational therapeutic effects of the cancer cell membrane-coated MOFs in cancer therapy. Similarly, Li et al. [75] devised tirapazamine (TPZ)-loaded Zr-based MOFs (PCN-224) coated with 4T1 cancer cell membrane for tumor-targeted PDT and hypoxia amplified bioreductive therapy. Cancer cell membrane coating allowed homotypic cancer-targeting ability, immune evasion, and selective tumor accumulation. PCN-224 produced cytotoxic ROS under visible light irradiation for PDT. In addition, the photochemical oxygen consumption induced hypoxic aggravation further facilitated the activation of TPZ for bioreductive chemotherapy. Noteworthily, cell membrane cloaking of MOFs exhibited a negligible impact on their intracellular ROS generation capacity. Therefore, this platform suppressed tumor growth and distal metastasis of breast cancers in a 4T1 tumor-bearing mice model.

### 4.2. Vascular-Related Diseases

Vascular disease is a pathological state of large and medium muscular arteries and is triggered by endothelial cell dysfunction. The stromal cell-derived factor-1α (SDF-1α)/CXC chemokine receptor 4 (CXCR4) axis is believed to play an important role in recruiting progenitor cells into ischemic tissue [114]. For example, Ma et al. [12] used CXCR4-overexpressed NSC membrane-coated PLGA nanoparticles for the delivery of glyburide for stroke treatment. The CXCR4-overexpressed cell membrane significantly enhanced the delivery of nanoparticles to the ischemic brain after intravenous administration, and it accumulated in the ischemic regions by 2.7 times greater than that of the naked PLGA nanoparticles in mouse middle cerebral artery occlusion (MCAO) surgery in vivo. Furthermore, the glyburide delivery through this nanoparticle significantly improved the mouse survival and neurological scores, and reduced infarct volume by 58% compared to that of free glyburide delivery. Similarly, Bose et al. developed a targeted VEGF delivery system based on PLGA nanoparticles coated with CXCR4-overexpressed stem cell membranes for ischemic hindlimbs repair [81]. In comparison to that of native stem cell membrane-coated nanoparticles, CXCR4-expressed stem cell membrane-coated nanoparticles facilitated nanoparticle accumulation in ischemic tissues by two-fold, and subsequently released VEGF, led to enhanced blood reperfusion, and improved limb salvage in a hindlimb ischemia model of peripheral vascular disease. Recently, Li et al. [70] reported that RBC–REC hybrid membrane-coated nanoparticles could act as an antiangiogenic material for targeted treatment of choroidal neovascularization (CNV), which is the leading cause of vision loss in many blinding diseases. Due to the RBC membrane fraction, the nanoparticle phagocytosis by macrophages was reduced, leading to the enhanced accumulation in CNV regions rather than in the liver. More importantly, inheriting VEGF receptors (i.e., VEGFR1 and VEGFR2) of the REC membrane fraction acted as anti-VEGF nanoagents and adsorbed VEGF-A ligands, resulting in the blocking of their effects on host endothelial cells and ocular angiogenesis. Therefore, the leakage and damage area of CNV in eyes treated with RBC–REC hybrid membrane-coated nanoparticles were significantly reduced compared with those in eyes treated with either REC-coated or RBC-coated nanoparticles. Taken together, recent progress demonstrates greater therapeutic effects of engineered membrane-coated nanotherapeutics on vascular-related diseases than naïve cell membrane-coated ones.

### 4.3. Infection

Infection is caused by the invasion of pathogenic microorganisms such as bacteria, viruses, and fungi. Upon infection, host immune systems fight with toxins produced by the microorganisms. Antibiotics are the first-choice treatment option; however, the overuse and misuse of the medication often causes antibiotic resistance, and therefore becomes a serious public health problem. To address this issue, antibiotic delivery systems responsive to pathogen-associated enzymes (e.g., penicillin G amidase, β-lactamase, phosphatase, and phospholipase) or intracellular reducing conditions were developed and the on-demand antibiotic release reduced the risk of adverse effects and drug resistance [115]. Recently, inspired by the toxin neutralizing properties of RBC membranes, the on-demand antibiotic delivery strategy was combined with the toxin decoy strategy for targeted toxin neutralizing and antibiotic release against intracellular infection. In RBC membrane-coated nanotherapeutic platforms, a nanoparticle core is used for stimuli-responsive antimicrobial delivery and release of antibiotics, while the RBC membrane shell plays a role in adsorbing and neutralizing toxins secreted from bacteria [35,85]. Furthermore, as such nanoparticles formed colloidal gels by oppositely charged nanoparticles (e.g., positively charged chitosan-functionalized ones), the retention of the gel at the injection site was significantly enhanced, leading to excellent antivirulence therapeutic effects against local bacterial infections [85]. The duration of the gel was affected by multiple factors, including gel composition, nanosponge size, core degradation rate, disease types, and route of administration. More importantly, in a mouse model of subcutaneous group A Streptococcus (GAS) infection, the lesion size of infected skin treated with the gel was drastically reduced, while that of skin treated with free RBC membrane-coated nanoparticles showed no significant reduction, indicating synergy between cell membrane-coated nanotherapeutics and gel-like bulk assembly formation. Such a decoy strategy was used for the treatment and prevention of viral infection as well. PLGA nanoparticles coated with CD4^+^ T cell membrane could bind to HIV via human CD4 receptor and CCR5 or CXCR4 co-receptors presented on the surface, resulting in inhibiting viral entry or fusion and intercepting HIV pathogenesis [86]. 

### 4.4. Inflammation

Inflammation is a protective response involving immune cells, and a molecular mediator directed against harmful stimuli and is closely associated with many human diseases. A variety of pro-inflammatory cytokines (e.g., interleukin-1 beta (IL-1b), tumor necrosis factor-alpha (TNF-a), and IL-6) [116,117,118] are known to play prominent roles in the progression of inflammatory diseases such as rheumatoid arthritis, pneumonia, and gout. Immune cell membranes (e.g., neutrophils and macrophages) are often used for inflammation-targeted therapy [119]. It was reported that local injection of neutrophil membrane-coated nanoparticles at the knee joint of a rheumatoid arthritis model could reduce overall arthritis progression and severity at the systemic level by actively neutralizing diffusive arthritogenic factors at the knee joint. As compared to that of RBC membrane-coated nanoparticles, neutrophil membrane-coated ones showed enhanced cartilage penetration due to the adhesion interactions between neutrophils and chondrocytes [84]. Similarly, macrophage membrane-coated MOFs (MΦ-MOF) were demonstrated to show greater neutralization of proinflammatory cytokines as well as inflammation targeting ability as compared with RBC membrane-coated MOFs [11]. Uricase, an enzyme that digests uric acid into allantoin, was loaded into the MOF by a facile formulation process with precise input control. As MΦ-MOF delivered uricase to the joint inflammatory sites, uric acid deposited in the joint was significantly degraded. Mice treated with uricase-loaded MΦ-MOF showed markedly reduced immune cell infiltration in the periarticular ankle tissue and alleviated the ankle swelling 48 h after treatment, demonstrating macrophage membrane coating of nanoparticles synergized with uricase delivery for hyperuricemia and gout treatment. In a murine model of endotoxin-induced lung inflammation, nanoparticles coated with cell membrane genetically modified to express VLA-4 could specifically target inflamed endothelial cells in the inflamed lung, and subsequent release of dexamethasone, an anti-inflammatory drug, significantly reduced inflammatory cytokine (i.e., IL-6) levels [14]. However, free dexamethasone treatment or dexamethasone delivery via wild-type cell membrane-coated nanoparticles showed negligible therapeutic effects. Taken together, these studies demonstrate that cell membrane cloaking of nanoparticles leverages the targeted drug delivery in inflammation therapy.

## 5. Perspectives for Clinical Translation

Although cell membrane cloaking techniques hold promise in biomedical applications, there are still technical obstacles limiting the clinical translation of cell membrane-based nanotherapeutics. Many interfacial aspects of nanoparticles affect completeness of membrane coverage and membrane sidedness, often leads to the partial coating [120,121]. Although partially coated nanoparticles could still be internalized by the target cells, standardizing protocols of cell membrane isolation and coating is essential to achieve consistent therapeutic effects of cell membrane-cloaked nanotherapeutics for clinical translation. Given that cell membrane-based nanotherapeutics are suitable for personalized medicine, autologous cell sources will be increasingly needed for a more practical approach and patient-specific disease treatment. Banking of autologous cells or cell membranes, and allogeneic materials from type-matched donors is crucial to expand the use into clinical practice. In the view of manufacturing of the cell membrane-based nanotherapeutics for clinical translation, standardized protocols for large-scale cell culture and cell membrane isolation need to be established. Moreover, proper quality controls for cell engineering and scale-up manufacturing of cell membrane-based nanotherapeutics would be required to avoid any safety concerns. Long-term storage methods for cell membrane-based nanotherapeutics are essential to offer clinically relevant off-the-shelf platforms in a relatively inexpensive, stable, and reproducible way. 

From a wider perspective, it appears inevitable that cell membrane cloaking technology is combined with various biomaterials such as scaffold-based therapeutic platforms which can expedite clinical use and improve the accessibility of disease treatments. Moreover, the cell membrane was demonstrated as coating materials for biointerfacing, which can lead to new functional nanomaterials or nanodevices for disease treatment and diagnosis [52]. For example, Chen et al. [122] demonstrated that pancreatic beta-cell membrane-coated nanofiber scaffolds enhanced cell proliferation and insulin secretion function in vitro. Recently, we reported a scaffold-mediated CRISPR/Cas9 local delivery system for acute myeloid leukemia therapy [123]. Mesenchymal stem cell membrane-coated nanofiber scaffolds enabled leukemia stem cell targeting as they were injected in the bone marrow. Noteworthy, the scaffold-based delivery increased the retention time of CRISPR/Cas9 in the injection site as compared to the nanoparticle-based delivery, leading to reduced long-term leukemic burden in vivo. Similarly, Zhang et al. [85] demonstrated that colloidal gels formed with negatively charged RBC membrane-coated nanoparticles and positively charged chitosan-functionalized nanoparticles showed superior antivirulence therapeutic effects to free RBC membrane-coated nanoparticles due to the prolonged retention of the gel at the injection site. This study also indicates that material diversity and formulation flexibility make the biomedical approach of cell membrane-based nanotherapeutics broadly applicable. As researchers continue to refine existing therapeutic platforms combined with cell membrane cloaking technology, there is still significant room for development.

## Figures and Tables

**Figure 1 ijms-23-02223-f001:**
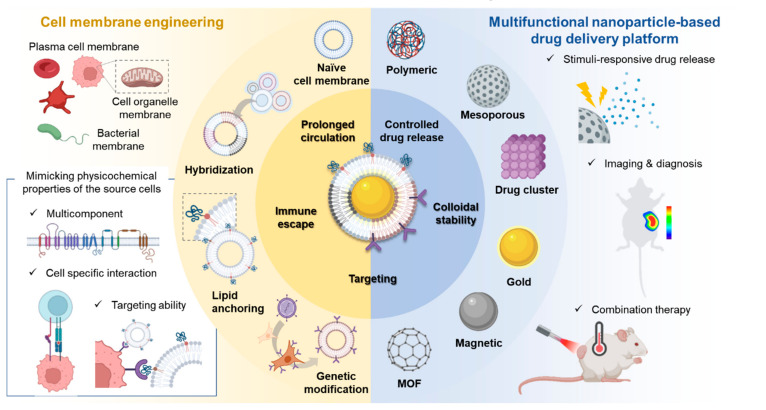
Schematic overview of cell membrane-cloaked nanotherapeutics for targeted drug delivery. (**Left**) plasma cell membranes, cell organelle membranes, bacterial membranes are often used as a coating material for nanoparticles to mimic physicochemical properties of source cells, for example, multicomponent, cell-specific interaction, and targeting ability. Recent advances in cell membrane engineering (e.g., hybridization, lipid anchoring, and genetic modification) allow cell membrane-based nanotherapeutics to achieve even more diverse and improved targeting effects along with prolonged circulation and immune escape. (**Right**) various nanoparticles including polymeric, mesoporous, drug cluster, gold, magnetic, and metal-organic framework are used as a core for cell membrane-based nanotherapeutics. Multifunctional nanoparticle-based drug delivery platforms allow stimuli-responsive drug release, imaging, diagnosis, and combination therapy in biomedical field. Figure created with BioRender.

**Table 1 ijms-23-02223-t001:** Unique features of natural cell membranes.

Cell Type		Features
Plasma membrane	RBC	Immunosuppressive effect, long-term blood circulation, and neutralization of toxins
	Platelet	Specific binding affinity to injured vasculature, pathogen adhesion, reduced cellular uptake by macrophages, and prolonged blood retention time
	Macrophage	Inflammation site-specific accumulation, neutralization of inflammatory cytokines, antigen-homing affinity
	Cancer	Homotypic targeting, and low intrinsic immunogenicity
Cell organelle membrane	Mitochondria	Selectively binding to mitochondrial membrane ligands, and neutralization of toxins
	Nucleus	Improved transfection efficiency in gene therapy

**Table 2 ijms-23-02223-t002:** Summary of cell membrane-based nanotherapeutics for targeted drug delivery.

Target Disease	Cell Membrane Isolated from	Core Nanoparticle	Drug/Surface Modification	Strategy	Ref
Cancer					
Breast cancer	MCF-7	MOF	CRISPR/Cas9	Homologous tumor targeting	[1]
	MDA-MB-231	Rare-earth doped nanoparticle	-	Homologous tumor targeting Tumor imaging in NIR-II window	[5]
	4T1	MnO_2_-coated MOF	Apatinib	Homologous tumor targeting Introducing photosensitive porphyrinic into MOF for enhanced PDT	[6]
	RBC and MCF-7	Melanin nanoparticle	-	Prolonged circulation half-life and homotypic tumor targeting Generating hyperthermia to increase PTT efficacy	[2]
	Fibroblasts activated with TGF-β1	Semiconducting polymer nanoparticle		Targeting cancer-associated fibroblasts Generating NIR fluorescence and photoacoustic signals for imaging Generating singlet oxygen and heat for combined PDT and PTT	[7]
	MCF-7	MSN	DOX and MPH	Homologous tumor targeting High drug loading	[19]
	RBC	Magnetic MSN	HB	Prolonged circulation High drug loading Tumor accumulation via magnetic navigation for improved PDT	[3]
	RBC	Semiconducting polymer nanoparticle	-	Prolonged circulation Photoacoustic imaging and PTT	[74]
	Macrophage	pH-sensitive nanoparticle	PTX/IGF1R-targeting ligand	Membrane-derived tumor homing Improved intracellular uptake by decorated with the IGF1R targeting ligand H^+^-adsorbing proton sponge effect accelerating endosomal escape of the nanoparticle Controlled drug release in the acidic intracellular environment	[42]
	4T1	MOF	TPZ	Homologous tumor targeting High drug loading in porous coordination network of MOF ROS generation under light irradiation	[75]
	RBC	MOF	ICG and oxygen	Prolonged circulation Facilitating O_2_ release from MOF by converting NIR light into heat O_2_-evolving PDT	[4]
	Natural killer cell	PLGA nanoparticle	TCPP	Tumor targeting via interactions between NKG2D and DNAX accessory molecule 1 Photosensitizer delivery for improved PDT Cascade immunotherapy	[76]
	Platelet	MOF	Survivin siRNA	Tumor targeting High siRNA loading and minimal toxicity	[17]
Cervical carcinoma	RBC	PEG-b-PDLLA nanoparticle	PTX dimer and TPC	Prolonged circulation Generating ROS under light irradiation for PDT and for triggering on-demand PTX release for chemotherapy	[77]
Head and neck squamous cell carcinoma (HNSCC)	HNSCC patient-derived tumor cell	Gelatin nanoparticle	Cisplatin (Pt)	Homologous tumor targeting	[78]
	Platelet and NHSCC cancer stem cell	Iron oxide nanoparticle	-	Homologous tumor targeting of cancer stem cell membrane Immune evasion of platelet membrane Optical adsorption ability and magnetic properties for PTT and MRI	[20]
Oral squamous cancer	KB	Gold nanorod		Homologous tumor targeting Perinuclear accumulation Combination of photothermal therapy and radiotherapy	[79]
Melanoma	CD80-overexpressing B16	PLGA nanoparticle	-	Promoting activation of the cognate T cells	[13]
	OVA-expressing B16	PLGA nanoparticle	R837/Mannose modification	Vaccination with enhanced intracellular uptake by antigen-presenting cells by mannose modification Checkpoint blockade therapy	[15]
	Myeloid-derived suppressor cell	Iron oxide nanoparticle		Homologous tumor targeting and immune escape PTT-induced tumor-killing PTT-mediated antitumor response	[21]
Glioma	RBC	Drug crystal	DTX/modified with tumor-targeting peptide c	High drug loading Prolonged blood circulation Active tumor targeting	[80]
Colorectal carcinoma	RBC	MOF	GOx, TPZ	Prolonged blood circulation Tumor hypoxia by GOx-based starvation therapy Starvation-activated TPZ activation	[22]
Vascular related disease					
Choroidal neovascularization (CNV)	RBC-REC	PLGA nanoparticle	-	Reducing phagocytosis by macrophages using RBC membranes Improved accumulation in CNV regions using REC membranes Neutralizing VEGF by inheriting VEGF receptors of REC membranes	[70]
Peripheral vessel disease (PVD)	CXCR4-overexpressed hASC	PLGA nanoparticle	VEGF	Reduced phagocytosis and promoted penetration across inflamed endothelial barrier using engineered cell membrane Targeted VEGF delivery to ischemic injury	[81]
Stroke	CXCR4-overexpressed NSC	PLGA nanoparticle	Glyburide	Chemotactic interaction with SDF-1, enriched in the ischemic microenvironment Targeted delivery of the anti-edema agent, glyburide for stroke treatment	[12]
Atherosclerosis	Platelet	PLGA nanoparticle	Gadolinium	Atherosclerosis targeting Live detection of atherosclerotic sites by MRI imaging	[82]
	Platelet	PAAO-UCNP	Ce6 photosensitizer	Atherosclerosis targeting ROS-induced apoptosis by SPECT/CT-guided PDT	[41]
	RAW 264.7	PLGA nanoparticle	Rapamycin	Inhibiting phagocytosis Atherosclerosis-targeted drug delivery	[83]
Inflammation					
Lung inflammation	VLA-4-expressed leukemia cell	PLGA nanoparticle	DEX	Enhanced affinity to target inflamed endothelial cells via VCAM-1 and VLA-4 interaction Anti-inflammatory drug delivery to inflamed sites	[14]
Gout	Macrophage	MOF	Uricase	High-yield enzyme loading Inflammation-targeted enzyme delivery Inflammatory cytokine-neutralization	[11]
Rheumatoid arthritis	Neutrophil	PLGA nanoparticle		Decoying neutrophil-targeted biological molecules Neutralizing pro-inflammatory cytokines Increased penetration into the cartilage matrix	[84]
Infection					
Skin infection by streptococcus	RBC	PLGA and chitosan nanoparticles		Toxin neutralization Prolonged retention by forming a gel-like complex	[85]
Methicillin-resistant staphylococcus aureus infection	RBC	pH-sensitive nanogel	Vancomycin	Toxin neutralization Redox-responsive antibiotics delivery	[35]
Bacterial infectious disease	E. coli	Gold nanoparticle		Anti-bacterial vaccination Targeted activation of dendritic cells in lymph nodes, triggering subsequent immune responses	[60]
Human immunodeficiency virus infection	SUP-T1, a human T lymphoblast cell line	PLGA nanoparticle		Viral targeting via CD4 receptor and CCR5 or CXCR4 coreceptors Decoying T cell-targeted virus, blocking viral entry and infection	[86]

MOF, metal-organic framework; RBC, red blood cell; NK, natural killer; ASC, adipose-derived stem cell; NSC, neural stem cell; PDT, photodynamic therapy; PTT, photothermal therapy; NIR, near-infrared; MSN, mesoporous nanoparticle; DOX, doxorubicin; MPH, mefuparib hydrochloride; HB, hypocrellin B; PTX, paclitaxel; TPZ, tirapazamine; ICG, indocyanine; TCPP, 4,4′,4′′,4′′′-(porphine-5,10,15,20-tetrayl) tetrakis (benzoic acid); PEG-b-PDLLA, methoxypoly(ethylene glycol)-block-poly(D,L-lactide); OVA, ovalbumin; DTX, docetaxel; GOx, glucose oxidase; REC, retinal endotheliocyte; VEGF, vascular endothelial growth factor; CXCR4, C-X-C Motif Chemokine Receptor 4; SDF-1, stromal cell derived factor 1; PAAO-UCNP, lanthanide-doped upconversion nanoparticles (UCNPs) incorporated into polyacrylic acid-n-octylamine (PAAO) micelles; ROS, reactive oxygen species; SPECT/CT, single-photon emission computed tomography/computed tomography; VLA-4, very late antigen-4; DEX, dexamethasone; VCAM-1, vascular cell adhesion molecule 1; CCR5, C-C chemokine receptor type 5.

## Data Availability

Data are available within the article.

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
