# Peer review of "Cell Membrane-Cloaked Nanotherapeutics for Targeted Drug Delivery"

_ijms, 2022, doi:10.3390/ijms23042223_

Round 1

Reviewer 1 Report

The paper presents very interesting review about cell membrane-cloaked nanotherapeutics for targeted drug delivery. It is a well written review, which presents in details the newest information  about cell membrane-cloaked nanotherapeutics. This paper is ready to publish in Int. J. Mol. Sci. in the present form.

Author Response

We sincerely appreciate the review for the comment. We have carefully corrected typos and grammatical errors and all changes are noted in highlighted text in the revised manuscript.

Reviewer 2 Report

1) Replicating the membrane properties (e.g., biochemical, mechanical) is an idea scenario, which is unlikely to be achieved in practice. Reviewers are asked to modify the relevant sections (abstract, section 2).

2) Quality of coating (e.g., uniformity) is an important consideration. This issue must be included.  Please see and cite: L. Liu et al, Nature Communications, 5726, 2021, DOI: 10.1038/s41467-021-26052-x.

3) Figure 1 needs a more descriptive caption to be more easily understood by the readers. Looks like the right side of the Figure has three sections. Form right to left: applications, substrate types; and then not clear what the other part (genetic modification, lipid anchoring, hybridization, naïve cell membrane) is supposed to represent.

4) It needs to be emphasized that features like prolonged circulation etc. all depend on the quality of coating and the actual (vs ideal) properties of the coating.

5) There is a recent review article related to RBC-based optical materials that needs to be cite: T. Hanley et al, Biomolecules, 11, 729, 2021,  DOI: 10.3390/biom11050729.

6) A Table should be added to compare the features, advantages and disadvantages of each cell type (RBC, platelet, macrophages) membrane.

7) The first report of ICG loaded into a nano-sized RBC-derived construct was by B. Bahmani et al., 2180, Scientific Reports, 2013,  DOI: 10.3390/biom11050729. This paper must be cited.

Other literature related to laser irradiation applications, image-guided therapy, and vascular and tumor imaging to be cited:

J. M. Burns et al, ACS Applied Materials and Interfaces, 10, 27621-27630, 2018.

X. Ren et al. Biomaterials, 92, 13-24, 2016.

S. Ye, et al, ACS Applied Materials and Interfaces, 11, 15262-15275, 2019.

W. Chen et al, Advanced Functional Materials, 27, 1605795, 2017, DOI: 10.1002/adfm.201605795

W. Jia et al, ACS Applied Materials and Interfaces, 12, 275-287, 2020.

J. M. Burns et al, Cancers, 13, 2544, 2021. DOI: 10.3390/cancers13112544.

8) References need to be formatted properly, including the use of appropriate abbreviations for journal names. Some citations are missing the journal name (e.g., reference 106).

Round 2

Reviewer 2 Report

Authors have addressed previous concerns.  I am happy to recommend this manuscript four publication.